# Fungal Elicitation Enhances Vincristine and Vinblastine Yield in the Embryogenic Tissues of *Catharanthus roseus*

**DOI:** 10.3390/plants12193373

**Published:** 2023-09-25

**Authors:** Dipti Tonk, Abdul Mujib, Mehpara Maqsood, Mir Khusrau, Ali Alsughayyir, Yaser Hassan Dewir

**Affiliations:** 1Cellular Differentiation and Molecular Genetics Section, Department of Botany, Jamia Hamdard, New Delhi 110062, India; dipti10785@gmail.com; 2Department of Botany, Government College for Women, M.A. Road, Srinagar 190001, India; meh_heem@yahoo.co.in; 3Department of Botany, Government Degree College (Boys), Anantnag 231213, India; mirkhusrau@gmail.com; 4Department of Plant and Soil Sciences, Mississippi State University, 75 B.S. Hood Rd, Starkville, MS 39762, USA; aa2942@msstate.edu; 5Plant Production Department, College of Food and Agriculture Sciences, King Saud University, Riyadh 11451, Saudi Arabia; ydewir@ksu.edu.sa

**Keywords:** callus induction, elicitation, Madagascar periwinkle, somatic embryogenesis, vincristine, vinblastine

## Abstract

Fungal elicitation could improve the secondary metabolite contents of in vitro cultures. Herein, we report the effect of *Fusarium oxysporum* on vinblastine and vincristine alkaloid yields in *Catharanthus roseus* embryos. The study revealed increased yields of vinblastine and vincristine in *Catharanthus* tissues. Different concentrations, i.e., 0.05% (T1), 0.15% (T2), 0.25% (T3), and 0.35% (T4), of an *F. oxysporum* extract were applied to a solid MS medium in addition to a control (T0). Embryogenic calli were formed from the hypocotyl explants of germinating seedlings, and the tissues were exposed to *Fusarium* extract elicitation. The administration of the *F. oxysporum* extract improved the growth of the callus biomass, which later differentiated into embryos, and the maximum induction of somatic embryos was noted T2 concentration (102.69/callus mass). A biochemical analysis revealed extra accumulations of sugar, protein, and proline in the fungus-elicitated cultivating tissues. The somatic embryos germinated into plantlets on full-strength MS medium supplemented with 2.24 µM of BA. The germination rate of the embryos and the shoot and root lengths of the embryos were high at low doses of the *Fusarium* treatment. The yields of vinblastine and vincristine were measured in different treated tissues via high-pressure thin-layer chromatography (HPTLC). The yield of vinblastine was high in mature (45-day old) embryos (1.229 µg g^−1^ dry weight), which were further enriched (1.267 µg g^−1^ dry weight) via the *F. oxysporum*-elicitated treatment, especially at the T2 concentration. Compared to vinblastine, the vincristine content was low, with a maximum of 0.307 µg g^−1^ dry weight following the addition of the *F. oxysporum* treatment. The highest and increased yields of vinblastine and vincristine, 7.88 and 15.50%, were noted in *F. oxysporum*-amended tissues. The maturated and germinating somatic embryos had high levels of SOD activity, and upon the addition of the fungal extracts, the enzyme’s activity was further elevated, indicating that the tissues experienced cellular stress which yielded increased levels of vinblastine and vincristine following the T2/T1 treatments. The improvement in the yields of these alkaloids could augment cancer healthcare treatments, making them easy, accessible, and inexpensive.

## 1. Introduction

*Catharanthus roseus*, commonly known as Madagascar periwinkle, is a tropical, perennial, medicinal plant belonging to the family Apocynaceae. It is a source of several important indole alkaloids of medicinal importance such as vinblastine, vincristine, ajmalicine, vindoline, catharanthine, and serpentine [1]. Due to its immense pharmaceutical importance and low (0.0005%) contents of vinblastine and vincristine, *C. roseus* has been regarded as an important model for secondary metabolism studies. In recent decades, an inclusive, multidimensional research study has attempted to improve the alkaloid contents in *C. roseus* [2,3]. Birat et al. [4] recently reported that the fungus *Nigrospora zimmermanii*, which is present within the leaves of *Catharanthus roseus*, also produced vincristine successfully. The strategies frequently used for enriching the levels of alkaloids are the optimization of media, plant growth regulators, and cultural practices; the culture of high-yielding cell lines; the use of precursors; the incorporation of elicitors; and improving the expression of the regulatory enzymes of metabolic pathways [5,6,7].

In recent years, researchers have attempted to influence the production of secondary metabolites from diverse tissue sources through the use of different biotic and abiotic elicitors [8]. The techniques used drastically reduced the processing times needed to obtain active compounds [9,10]. Elicitors are a large target group of compounds which have been added to media at various stages of cultural growth for improving secondary compounds. Traditionally, an ‘elicitor’ is a molecule which is introduced into a medium in small levels to improve the biosynthesis of compounds by triggering cellular defense response genes [11,12]. The process also refers to compounds of various sources which stimulate physiological and morphological responses in inducing compounds of a defensive nature [13]. It is well established that the application of an elicitor or the invasion of a pathogen produces an array of defensive secondary reactions in plant cells. Singh et al. [14] categorized diverse types of elicitors diverse types: (a) biotic elicitors such as bacterial and fungal cell walls or glycoproteins, (b) abiotic elicitors like UV irradiation, salt, and various non-constitutive compounds, and (c) endogenous elicitors, which are signaling compounds of plant-cell origins. A large number of biotic elicitors have been recognized to be very efficient at enriching secondary metabolites and are exploited in a variety of cultures [15]. Yeast extract was used as a biotic elicitor in cultures which induced the synthesis of a variety of phytocompounds in several investigations into plant–microbe interactions [16,17]. Endophytic fungi (used as fungal elicitors) isolated from *C. roseus* could also be used to enrich indole alkaloid production in culture [18]. The culture filtrate of *Fusarium sporotrichioides* Sherbakoff, isolated from *Narcissus tazetta var. italicus* rhizosphere and grown on a potato dextrose broth, stimulated the production of alkaloids in cultivated tissues [19]. A marked increase in vasicine content in *Adhatoda vasica* was observed via the amendment of select elicitors like methyl jasmonate (MeJA), chitosan, yeast extract, ascorbic acid, and sodium salicylate at optimized concentrations [20]. Arbuscular mycorrhizal fungi (a group of beneficial microorganisms) were reported to play a major role in enhancing alkaloid production in root organ cultures [21]. In *Centella asiatica*, the influence of various elicitors, like the use of *Trichoderma harzianum*, *Colletotrichum lindemuthianum*, and *Fusarium oxysporum* to improve the accumulation of secondary metabolites, was reviewed and discussed [22,23].

In addition, a number of abiotic factors have been widely incorporated to augment product synthesis in cultured tissues, such as elevated temperature, excess salinity, osmotic stress, ultra-violet (UV) rays, and heavy metal stress [24]. In this specific plant, *C. roseus*, a variety of abiotic compounds such as NaCl, cerium (CeO_2_ and CeCl_3_), yttrium (Y_2_O_3_), and neodymium (NdCl_3_) were used successfully to enhance alkaloid yield [25]. CaCl_2_ was used as an elicitor for the enhancement of vinblastine in a *C. roseus* embryogenic cell suspension [26]. When used, these elicitors caused stresses and improved the synthesis of secondary compounds in several investigated genera. Elicitor-induced cellular stress is measured by monitoring antioxidant enzymes, which ameliorate stresses in cultures [27,28]. Various enzymes such as superoxide dismutase (SOD), catalase (CAT), ascorbate peroxidase (APX), and glutathione reductase (GR) are assayed to ascertain the level of stress in cultured tissues and were studied in different plant genera [29]. Although the enhancement of alkaloids is noted to be treatment-specific, the use of elicitors could be a valuable strategy for enriching phytocompounds. In this study, the fungus *Fusarium oxysporum* was used as biotic elicitor, and the yields of vinblastine and vincristine were measured in cultures. This is perhaps one of the first fungal (biotic) elicitation studies on alkaloid yield mediated via embryogenesis. The growth of the callus biomass and the biochemical alterations/associations during the course of its growth and morphogenesis were monitored.

## 2. Results

### 2.1. Callus Induction and Biomass Growth in a Medium Containing a Fusarium oxysporum Extract

On MS medium supplemented with 4.52 µM of 2,4-D, the hypocotyls of seedlings grown in vitro produced profuse calli. The calli were friable, light-yellow, and fast-growing; they later turned into embryogenic calli (Figure 1a). These hypocotyl calli were subjected to various levels of *F. oxysporum* elicitation and routinely subcultured at regular intervals. Growth is an indicator of cell division with the rapid multiplication of the callus; therefore, the growth the biomass of each callus was measured in response to the elicitor treatments. We observed that with the *F. oxysporum* treatment, the growth of the embryogenic calli was faster compared to the control. The biomass of each calli increased up to T2, and for this treatment, maximum fresh, dry, and absolute dry mass % values were observed (1.55, 0.183 g, and 11.803%, respectively). Upon elicitation, the calli appeared friable and white, especially those that received the T1 and T2 treatments. The calli that received higher concentrations, i.e., T3 and T4, were less responsive; the calli turned light-brown, were compact, and showed poor growth.

### 2.2. F. oxysporum Treatments and the Number of Embryos 

The embryogenic calli were cultured on MS medium supplemented with 5.37 µM of NAA and 6.72 µM of BA, and different concentrations of the *F. oxysporum* extract were added in order to monitor the influence of the fungus elicitor on the number of embryos and their growth. The maximum fresh, dry, and absolute dry weight values were observed for T2 (2.066, 0.237 g, and 11.442%, respectively) compared to the other treatments and the control, T0. Under all the tested conditions, the embryogenic calli differentiated into embryos, and for theT2 concentration of the *F. oxysporum* treatment, the maximum number of embryos was formed (102.69/culture) (Table 1). The next important treatment was T1 (94.36/culture), which induced a good number of embryos; the embryo numbers declined gradually at higher elicitor levels.

### 2.3. The Maturation and Germination of Somatic Embryos in a Medium Containing F. oxysporum

The cotyledonary embryos were cultured on MS medium supplemented with 2.60 µM of GA_3_ for maturation; the medium was additionally supplemented with the fungal elicitors (Figure 1b). For the concentrations T1 and T2, the embryos were elongated, coiled, and turned green, and they later germinated into plantlets (Figure 1c). For the concentrations T3 and T4, however, embryo development was poor; a few remained in an advanced cotyledonary stage, while the others turned brown. The embryos that reached maturity were thin and showed poor growth. The embryos germinated into plantlets on the MS medium containing 2.24 µM of BA. The percent germination and the shoot and root lengths of the germinated somatic embryos were higher under the *F. oxysporum*-elicitated conditions compared to the control (Table 2).

### 2.4. Vinblastine and Vincristine Yields

The yields of vinblastine and vincristine were quantified in different in vitro-cultivated tissues. The mobile phase showed sharp standard vinblastine and vincristine peaks. A regression analysis also showed a good linearity, with r = 0.999 and 0.993 for vinblastine and vincristine, respectively. It is evident from Table 3 that the maximum yields of vinblastine were achieved in the embryos’ maturation (0.788 µg g^−1^ dry weight; Figure 2a,c) and germination stages (0.835 µg g^−1^ dry weight; Figure 2b,d) compared to the other two stages, i.e., the induction and proliferation stages of the embryo tissues. With *F. oxysporum* elicitation at T2, the vinblastine yield was further improved (0.886 µg g^−1^ dry weight), and the T1 treatment was equally efficient in promoting its yield. Compared to vinblastine, the yield of vincristine was low, and the maximum content was achieved in germinating embryos compared to the other stages. Upon the addition of *F. oxysporum*, an improved vincristine yield was noted in the cultured tissues (Table 4), with the maximum identified for the T2 treatment (0.307 µg g^−1^ dry weight), followed by the T1 treatment (0.275 µg g^−1^ dry weight). The maximum increased yields of vinblastine and vincristine, 7.88 and 15.50%, respectively, were noted for the *F. oxysporum*-elicitated treatment T2 over the control tissues.

### 2.5. Fusarium Oxysporum Elicitation and Biochemical Attributes

#### 2.5.1. Sugar, Proline, and Protein Contents

As the *F. oxysporum* elicitation, especially at low levels, improved alkaloid yields, we attempted to monitor various non-enzymatic stress markers for different tissues. The sugar content was noted to be high during the embryos’ maturation stage compared to the germination stage. Upon the addition of increased levels of elicitors, the sugar level increased further, reaching a maximum in T2 (21.663 mg g^−1^). The proline level was also high in the maturation stage (8.255 mg g^−1^), but the proline accumulation declined with the growth and maturation of the embryos (7.254 mg g^−1^) at T2. The total soluble protein, on the other hand, was found to be more or less the same at these two advanced stages, i.e., maturation and germination,; comparative details of the elicitation doses and biochemical attributes are presented in Table 5 and Table 6.

#### 2.5.2. SOD, CAT, and APX Activities

The germinating and maturated somatic embryos showed enhanced levels of alkaloids, especially on the *F. oxysporum*-treated culture. The addition of an elicitor might cause stress for tissues. To better understand the impact of the elicitor treatments on plant defense and later on secondary metabolism, the antioxidant activities of various enzymes were investigated as stress markers. The maturated and germinating somatic embryos had higher levels of antioxidant enzyme activities than the early embryogenic tissues. The antioxidant enzyme activities were higher upon the addition of the *F. oxysporum* treatments, which indicated extra cellular stress on the cultivated tissues. It is evident from Figure 3 that at T_2,_ the levels of SOD activity were high in the maturation (4.115 EU min^−1^ mg^−1^ proteins) and germination (3.693 EU min^−1^ mg^−1^ proteins) stages of the embryos compared to the control (3.785 and 3.415 EU min^−1^ mg^−1^ proteins respectively), which yielded the highest levels of vinblastine and vincristine. Compared to SOD, the activities of CAT and APX were, however, low, i.e., 2.355 and 1.075 min^−1^ mg^−1^ protein, respectively, in the embryos’ maturation stage. The germinating somatic embryos also had similarly low levels of CAT and APX enzyme activity (Figure 4).

## 3. Discussion

In the present study, the yields of vinblastine and vincristine were quantified following *F. oxysporum* elicitation in embryogenic cultures of *C. roseus*. The callus was induced from hypocotyls on MS medium supplemented with 2,4-D in which-high frequency somatic embryos were formed; other auxins used induced embryos at a slower rate. Here, embryo differentiation was noted on the embryogenic callus, i.e., indirectly, but in other observed cases, embryos were also formed directly on explants without an intervening callus [30]. In both embryo-forming developmental pathways, the use of exogenous auxins/auxin analogues like 2,4-D efficiently promoted embryogenesis. These synthetic auxin analogs play a central signaling role in the acquisition of embryogenic competence from a somatic state [31,32]. In our study, an *F. oxysporum* extract was used at varying concentrations, of which T2 (0.15%) was observed to be more efficient at promoting callus biomass growth compared to T1, T3, and T4. We also observed that the callus biomass and the number of embryos increased significantly in T2 with *F. oxysporum* elicitation. The induced embryos were distinct and showed fast growth and development under the elicitated condition. The results of the present study indicate that the high concentrations (T3 and T4) of elicitation decreased the growth of the callus biomass by inhibiting cell division, and this reduction may have been due to the toxicity of the fungal extract or the excessive availability of stress ions [33]. In the present study, the addition of a low level of the *F. oxysporum* extract improved the number of somatic embryos in the culture. Similar responses, i.e., stress-induced embryogenesis, were described earlier in a number of previous observations [34,35]. Once an embryo is induced, the presence of 2,4-D in the medium inhibits the embryo’s development; therefore, other PGR combinations were tested and suggested to be necessary [35]. The involvement of cytokinins alone or with low doses of a weak auxin like NAA successfully influenced in vitro embryogenesis and plant morphogenesis [36,37].

The cultivation of plant cells and tissues or complex, organized structures is practiced in vitro as an efficient renewable source for the production of a variety of phytochemicals, and the importance of these methods were reviewed in recent years [38,39]. Calli and suspensions are cultivated more frequently because of their ease of cultivation and the possibility of scaling up their production in bioreactors. Aside from bioreactors, a number of other important strategies such as liquid culture, the use of mist, and liquid overlaying are used to improve biomass/embryogenesis to generate raw materials for alkaloid synthesis [40]. Liquid overlaying is a technique in which a thin film of a liquid nutrient is added on a solid medium to improve somatic embryogenesis in cultures [41]. The yields of active compounds are often high in complex, differentiated structures like shoots, roots, and leaves [14,42]. The method of extracting metabolites synthesized and accumulated in specialized cells or tissues is difficult, but genetically constructed biosensors can detect the precise locations of specialized metabolites at the tissue or cell level [43]. Different techniques have recently been adopted for the collection of alkaloids from specialized tissues. In the present study, we noted that compact embryo structures like maturated and germinating embryos synthesized higher yields of vinblastine and vincristine compared to embryos in early stages. Upon receiving *F. oxysporum* elicitation treatment, a 7.88% increased yield of vinblastine and a 15.50% increased yield of vincristine were noted. The same low level (T1/T2) of elicitation was noted earlier to be very efficient for improving the callus biomass. This rapid growth of the embryogenic callus may have been due to fast cell mitosis triggered by cell-cycle genes which were strongly upregulated in the dividing cells [44,45]. The influence of *F. oxysporum* on biochemical attributes was investigated as the addition of the elicitor improved the alkaloid yield. In the present study, extra sugar, protein, and proline accumulations were noted; however, these declined with increased levels of elicitation. Similar increases in protein, phenolics, hydrogen peroxide, and carbohydrates in response to stress were noted in several investigated plant genera, and these enhancements are considered good adaptation mechanisms in tolerant genotypes [46,47]. The protein level also increases gradually with the progress of tissues, and a change in protein with a progressing developmental stage was reported earlier in other investigated plant materials [42,48]. In tomato, enriched proline and lysine and glutamine accumulation were noted at an early stage of embryonic development, and this probably confers tolerance to drought [49]. Here, in the *F. oxysporum*-elicitated tissues, the increased accumulation of proline may have been due to the up-regulation of a proline synthesis gene which produced P5C reductase (PYCR) and proline dehydrogenase/oxidase (PRODH/POX) enzymes participating in the interconversion of intermediates in proline biosynthesis pathways [50,51]. Transcriptome data reveal that rice universally downregulates photosynthesis in response to abiotic and biotic stresses. At the same time, it also upregulates the hormone-responsive genes of the abscisic acid, jasmonic acid and salicylic acid pathways during stress [52]. In transgenic tobacco, the overexpression of AhCytb6 regulates the expression of various genes to enhance plant growth under a N_2_ deficit and abiotic stress conditions by modulating the plant’s physiology [53]. Enzymes like Cipk6, a Calcineurin B-like interacting protein kinase (CIPK) of tomato, regulates programmed cell death in immunity, transforming Ca^2+^ signaling in the formation of reactive oxygen species [54,55].

As the yields of alkaloids were high in the advanced-staged embryos, we tried to investigate the level of stress by measuring the activities of antioxidant enzymes in these cultivated tissues. The level of SOD activity was high in both of these two tissues, and upon the addition of the elicitor, the activity was further elevated. Increased SOD activity under various stresses was observed in several investigated plant genera [56,57]. CAT and APX also showed similar trends with added levels of elicitors, although tissue- and dose-specific variations were not uncommon [58,59]. In addition to the increases in the activities of stress marker enzymes and the alteration of physiological reserves, a molecular analysis indicated that the expression of the *Salt Overly Sensitive 1* (*SOS1*) gene is an important event in response to adaptive stress caused by biotic and abiotic factors [60]. It is very evident from the present study that the *F. oxysporum* elicitor promoted cultural growth in *C. roseus* and later stimulated enriched levels of alkaloids; however, the underlying mechanism is still not fully understood. It was reported earlier that the fungus extract in general contained compounds like sugars and proteins [61]. A chemical analysis showed that the hyphal walls of *F. oxysporum* are primarily composed of N-acetyl-glucosamine, glucose, mannose, galactose, uronic acid, and proteins or peptides [62,63]. The roles of various sugars, sugar alcohols, and related energy sources in improving synthesis were indicated earlier in several studies [64,65]. But the roles of protein or truncated proteins like small, moderate, or large peptides in triggering the synthesis of alkaloids have not been determined in a major way. Although the best mechanism of improving synthesis is not fully known, the process may be due to the formation of an ‘elicitor-receptor complex’ [66,67] which stimulates a cascade of defense genes in promoting alkaloid synthesis [68,69]. Thus, experimentations on elicitation through the use of various agents are immensely valuable as the technique promises to promote alkaloid biosynthesis in cultivated tissues.

## 4. Materials and Methods

The fruits/seeds of *Catharanthus roseus* (L.) G. Don were procured from the herbal garden of Jamia Hamdard (Hamdard University). The material was identified earlier, and a voucher specimen (JH-002-98) was maintained.

### 4.1. In Vitro Seed Germination and Culture Conditions

Seed germination and the process of establishing a culture of *C. roseus* L. (G). Don were carried out using the protocol established in our laboratory by [64]. In a nutshell, from twenty to twenty-five surface-disinfected seeds were placed in a 250 mL conical flask (Borosil, Mumbai, India) containing 50 mL of solid MS medium without any plant growth regulator (PGR). The germinated seedlings were maintained until the shoots attained a height of 2–4 cm. Various parts (the nodal stem, leaf, and hypocotyl) were used and inoculated in test tubes (Borosil, India) as explants. For the induction of an embryogenic callus, the MS medium was supplemented with 4.52 lM of 2,4-Dichlorophenoxyacetic acid (2,4-D). For the fast proliferation of embryos, the medium was fortified with 6.72 µM of N^6^-Benzyladenine (BA) and 5.37 µM of naphthalene acetic acid (NAA). All the above PGRs were procured from Sigma-Aldrich, St. Louis, MO, USA. The medium was solidified with 8 g L^−1^ of agar (Hi-media, Mumbai, India), and each tube contained 20 mL of medium. The pH of the medium was adjusted to 5.7 before it was autoclaved at 121 °C. All the cultures were incubated at 25 ± 2 °C under a 16 h photoperiod provided by cool-white fluorescent tubes at a photosynthetic photon flux density (PPFD) of 100 µmol m^−2^ s^−1^.

### 4.2. The Procurement and Culture of Fungi and the Preparation of the Elicitor

*Fusarium oxysporum* (Figure 5) was obtained from the Department of Pathology, Indian Agricultural Research Institute (IARI), Pusa, New Delhi, India. The fungus was grown in 100 mL conical flasks containing potato dextrose agar (Hi-media, India). After 7 d, the conical flasks containing fungal growth were sterilized and filtrated using Whatman no. 1 filter paper. The mycelium was washed several times with sterilized, distilled water and stored at 4 °C after being suspended in 100 mL water; this was designated as the culture media filtrate. The fungal mat was washed several times with sterilized, distilled water, and an aqueous extract was prepared [70] via homogenization with a mortar and pestle. This extract was filtered through centrifugation at 5000 rpm, and the supernatant was taken. It was later sterilized (designated as the mat extract) and kept at 4 °C for future investigations. Four different fungal elicitor treatments, i.e., 0.05% (T1), 0.15% (T2), 0.25% (T3), and 0.35% (T4), were prepared and added to the culture medium. A control (T0), i.e., a culture medium without the fungal filtrate, was also used for comparative evaluations of the elicitor’s influence. Morphogenetic and biochemical studies were conducted at periodic intervals.

### 4.3. Callus Induction under Fungus-Treated and Non-Treated Conditions

Hypocotyls of 5–6 d old seedlings were placed on MS and supplemented with an optimized 2,4-D concentration (4.52 µM). Four different treatments containing the *Fusarium oxysporum* fungal elicitor were added in order to assess the effect of the elicitors on callus induction and growth. A control, i.e., a medium without fungal filtrate, was also used for comparison. For a growth index analysis, callus biomass samples, i.e., the fresh and dry weights of calli at various growth stages, were taken and investigated. For the determination of the fresh weight, the calli (with or without elicitor treatment) were weighed immediately after isolation at regular intervals (15, 30, and 45 d). To determine the dry weight, the calli were dried at 60 °C for 18 h and measured, and the absolute dry mass was finally calculated using the method and formula of Winkelmann et al. (2004): Absolute dry mass (%) = Dry weight/fresh weight × 100.

### 4.4. The Proliferation, Maturation, and Germination of Embryos under the Influence of Biotic Elicitors

The embryogenic callus (40–50 mg) was cultured on MS supplemented with optimized concentrations of BAP (6.62 µM) and NAA (5.36 µM) for embryo proliferation. The medium was additionally amended with the above-mentioned fungus for the treatments indicated earlier treatments. The somatic embryos were induced in masses and were counted; this stage was called the proliferation stage. Vincristine and vinblastine alkaloids were extracted from the proliferation-stage embryos, and some of the proliferated embryos were cultured in medium for embryo maturation. The somatic embryos on MS supplemented with 2.89 µM of GA_3_ became larger and turned green, which is a good morphological indicator of matured embryos. The green, matured embryos were later placed on the same MS, supplemented with 2.22 µM of BAP for germination. The above two stages (maturation and germination) of the embryo development media were additionally supplemented with the *Fusarium oxysporum* extract for the above-indicated treatments. The somatic embryos started to germinate within a week or so, and the germination percentage and shoot and root lengths were measured and compared to assess the impact of the elicitor on the embryos. Matured and germinating embryos were harvested and oven-dried for the extraction of vincristine and vinblastine.

### 4.5. Vinblastine and Vincristine Quantification through HPTLC

Vinblastine and vincristine were extracted following methods described earlier methods [71,72] and their contents were measured in different in vitro-grown tissues and compared with standard vinblastine and vincristine obtained from Sigma-Aldrich (St. Luis, MO, USA). The selected tissues/embryos were collected from optimized media with their best growth. A total of 1 gm (dry weight) of tissues/embryos was refluxed in 30 mL of methanol for 5 h; later the supernatant was warmed at 60 °C, and the volume was finally reduced to 1–2 mL. Then, 1 mg of vinblastine and vincristine each was dissolved in 1.0 mL of methanol to make a stock solution concentration of 1.0 mg mL^−1^. Various concentrations were prepared from the stock solutions to obtain 200, 400, 600, 800, and 1000 µg per band of the standard and were assessed separately via HPTLC. A standard curve was plotted between the peak area (*y*-axis) and concentration (*x*-axis), which showed good linearity. For the stationary phase, thin-layer chromatography (TLC) aluminum sheets which measured 20 × 10 cm and were coated with silica gel (60 F 254, Merck, Bengaluru, India) were used. The freshly prepared mobile solution (phase) contained toluene, carbinol, acetone, and ammonia in a ratio of 40:20:80:2. The samples were applied using a 100 µL micro-syringe via a Linomat 3 (CAMAG) applicator. The silica plates were air-dried for 10–15 min and kept in a chamber (Twin Through Chamber CAMAG, 20 × 10 cm) filled with mobile solution. The solvent system was allowed to move up to about 85 mm. The plates were later removed from the chamber and air-dried again for about 10–20 min. The silica gel plates were documented using a CAMAG Reprostar under UV light without any chemical spray applied. The vinblastine- and vincristine-containing stationary phase was scanned via a CAMAG Scanner 3. The vinblastine and vincristine were scanned at 280 and 300 nm, respectively. The peaks of vinblastine and vincristine were fixed, and the identification of the alkaloids in the tissue samples was achieved by comparing the peaks of standard alkaloids. Finally, the alkaloid yields were measured in µg gm^−1^ of dry weight.

### 4.6. Estimation of Total Sugar, Proline, and Protein Contents

The estimation of the total sugar content was carried out according to the Dey method [73]. Tissues at different stages (0.5 g) were extracted twice with 90% ethanol (AR, New Delhi, India), and the extracts were pooled. The final volume of the pooled extract was increased to 25 mL via the addition of double-distilled water. To an aliquot of 1.0, 1.0 mL of 5.0% phenol and 5.0 mL of concentrated analytical-grade sulfuric acid were added and cooled in air. The optical density was measured at 485 nm. A solution containing 1.5 mL of 55% glycerol (AR, India), 0.5 mL of ninhydrin (AR, India), and 4.0 mL of double-distilled water was used as a calibration standard. For the measurement of proline, 0.2 g of specific stages of tissues were homogenized in 5.0 mL of 3% aqueous sulfosalicylic acid and filtered through Whatman filter paper (No. 1). To 1.0 mL of the extract, 1.0 mL of acid ninhydrin and 1.0 mL of glacial acetic acid (AR, India) were added, and the reaction mixture was incubated at 100 °C for 1 h. The reaction mixture was placed on ice and extracted using 2.0 mL of toluene. The proline content in the extract was subject to the spectrophotometric assay of Bates et al. [74]. The protein content was estimated via the Bradford method [75]; 0.5 g of tissue was ground in a pre-cooler mortar and pestle with 1.5 mL (0.1 M) of phosphate buffer (pH 7.0), placed on ice, and centrifuged at 5000 rpm for 10 min. With 0.5 mL of trichloroacetic acid (TCA), the sample was again centrifuged at 5000 rpm for 10 min. The supernatant was discarded, and the pellet was washed with chilled acetone and dissolved in 1.0 mL of 0.1 N sodium hydroxide (NaOH). Later, a 0.5 mL aliquot was added to 5.0 mL of Bradford reagent, and the optical density was measured at 595 nm.

### 4.7. Assay of Antioxidant Enzyme Activity

The catalase (CAT) activity was measured following the Aebi method [76]. It was measured by observing the decay in H_2_O_2_, and a decrease was measured at an absorbance of 240 nm in a reaction mixture containing 1.0 mL of a 0.5 M phosphate buffer (Na-phosphates, pH 7.5, AR, India), 0.1 mL of EDTA (AR, India), 0.2 mL of enzyme extract, and 0.1 mL of H_2_O_2_. The chemical reaction was continued for 3 min. The enzyme activity was represented as EU mg^−1^ protein min^−1^. A single unit of enzyme represents the amount used to decompose 1.0 µmol of H_2_O_2_/min. The enzyme activity was registered using the coefficient of absorbance at 0.036 mM^−1^ cm^−1^. The superoxide dismutase (SOD) activity was measured following the method of Dhindsa et al. [77]. Different stages of tissues/embryos (0.1 g) were homogenized in 2.0 mL of extraction solution (0.5 M of sodium phosphate buffer, pH 7.3, + 3.0 mM of EDTA + 1.0% (*w*/*v*) polyvinylpyrollidone (PVP, AR, India) + 1.0% (*v*/*v*) + Triton X100, AR, India), and the mixture was centrifuged (10,000 rpm) at 4 °C. The enzyme activity was measured by the ability to inhibit photochemical reduction. The assay mixture contained 1.5 mL of reaction buffer, 0.2 mL of methionine, 0.1 mL of enzyme extract, an equal amount of 1.0 M NaCO_3_ and 2.25 mM Nitro Blue Tetrazolium (NBT) solution, 3.0 mM of EDTA, riboflavin, and 1.0 mL of Millipore H_2_O. The whole mixture was kept in test tubes and incubated at 25 °C for 10 min under light. A 50% loss in color was considered 1.0 unit, and the enzyme content was expressed as EU mg^−1^ protein min^−1^. For ascorbate peroxidase (APX), the Nakano and Asada [78] method was used. The assay mixture contained 1.0 mL of 0.1 M sodium buffer, pH 7.2, + 0.1 mL pf EDTA + 0.1 mL of enzyme extract. The ascorbate was added to the solution and the reaction mixture was run for 3 min at 25 °C. The APX activity was measured by observing the reduction of absorbance by ascorbate mediated breakdown of APX. Enzyme activity was measured by using co-efficient of absorbance 2.81 mM^−1^ cm^−1^. Similar to other enzymes, the activity was expressed in EU mg^−1^ protein min^−1^ i.e., one unit of enzyme determines the amount necessary in decomposing 1.0 µm of ascorbate/min.

### 4.8. Statistical Analysis

The data on the effect of *Fusarium oxysporum* elicitor on callus growth and embryogenesis and differences in biochemical attributes, antioxidant enzyme activity, the alkaloid yield, and other parameters were analyzed via a one-way analysis of variance (ANOVA). The data or the values are the means of three replicates from two experiments and the presented mean values were separated using Duncan’s multiple range test (DMRT) at *p* ≤ 0.05.

## 5. Conclusions

Low doses of an *F. oxysporum* extract proved effective for improving callus biomass growth, embryogenesis, plant regeneration, and alkaloid yield in *C. roseus*. The percent germination and shoot and root lengths of somatic embryos were high at a low level (from 0.05% to 0.15%). Maturated and germinating somatic embryos had high levels of vinblastine and vincristine, which were further improved (to 7.8 and 15.5%) via elicitation. The addition of the elicitor caused cellular stress, which was evidenced by the biochemical attributes and high levels of antioxidant enzyme activities. We therefore recommend low doses of the fungal extract for enhancing the synthesis of alkaloids in *C. roseus*. The improvement in the yields of alkaloids could augment cancer healthcare in an easy and inexpensive manner.

## Figures and Tables

**Figure 1 plants-12-03373-f001:**
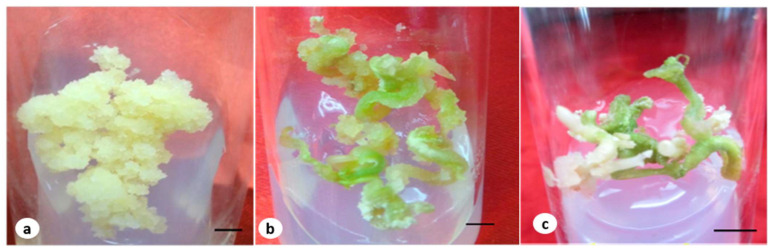
(**a**) Embryogenic callus grown in MS medium containing 4.52 µM of 2,4-D and the T2 fungal elicitor (bar 2 mm); (**b**) embryo on a maturation medium containing 2.60 µM of GA_3_ and an elicitor (bar 2 mm); (**c**) germinated embryos at early stage with a root (bar 0.5 cm).

**Figure 2 plants-12-03373-f002:**
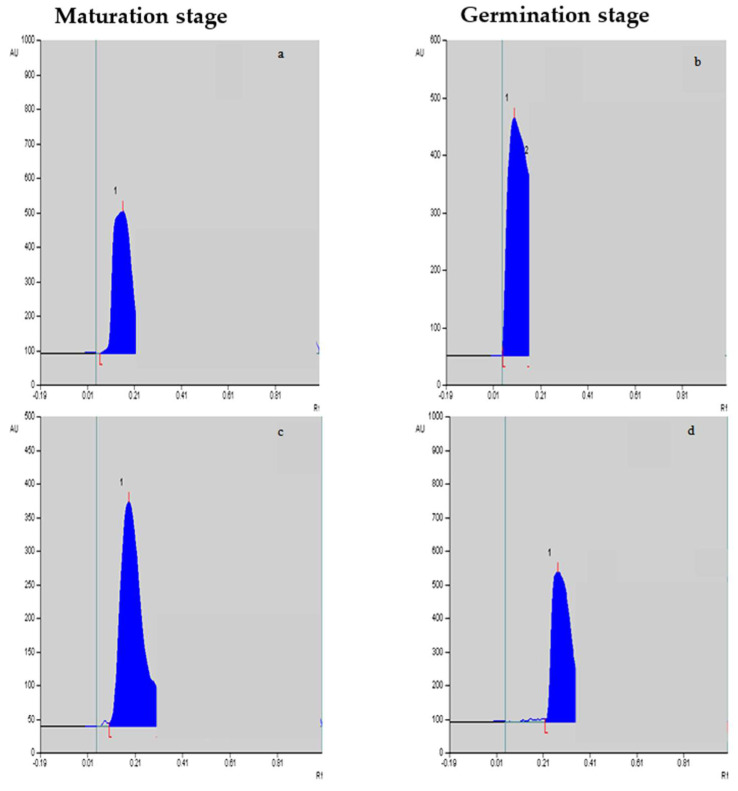
Vinblastine (**a**,**b**) and vincristine (**c**,**d**) peaks/levels at the maturation stage and germination stage, respectively, in response to *F. oxysporum* elicitation treatment at 0.25%.

**Figure 3 plants-12-03373-f003:**
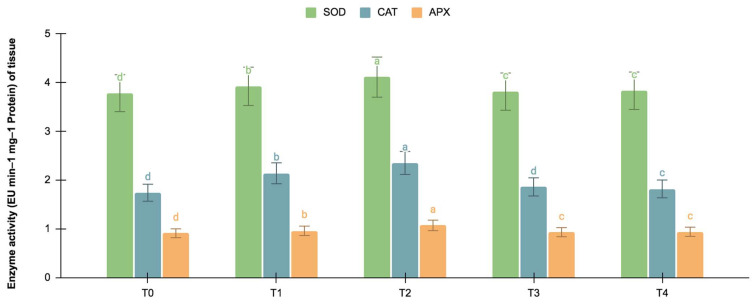
SOD, CAT, and APX activities (EU mg^−1^ protein min^−1^) in the maturation stage of the embryos for different *Fusarium oxysporum* treatments. The different *F. oxysporum* levels used were a control (T0), 0.05% (T1), 0.15% (T2), 0.25% (T3), and 0.35% (T4). The data were scored after 30 days of culture. The values are the means ± standard errors of three replicates. Within each column, the means with the letters are not significantly different at *p* ≤ 0.05, according to DMRT.

**Figure 4 plants-12-03373-f004:**
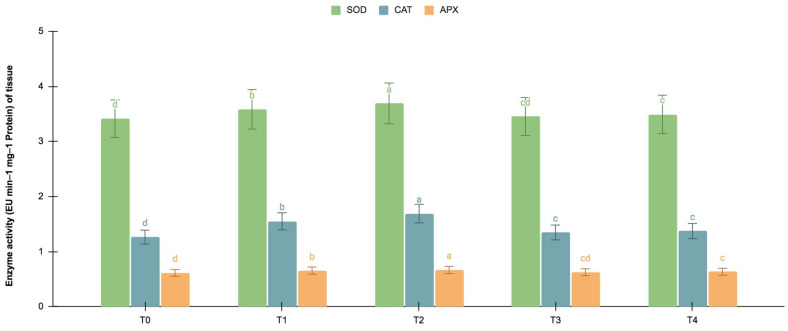
SOD, CAT and APX activities (EU mg^−1^ protein min^−1^) of the germination stage of the embryos for different *Fusarium oxysporum* treatments. The different *Fusarium oxysporum* levels used were a control (T0), 0.05% (T1), 0.15% (T2), 0.25% (T3), and 0.35% (T4). The data were scored after 30 days of culture. The values are the means ± standard errors of three replicates. Within each column, the means with the letters are not significantly different at *p* ≤ 0.05, according to DMRT.

**Figure 5 plants-12-03373-f005:**
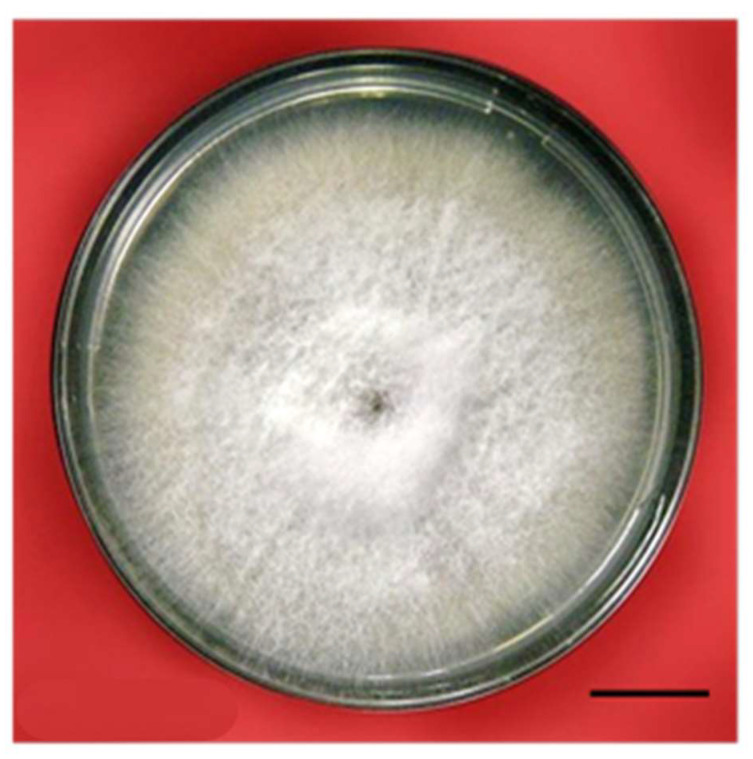
*Fusarium oxysporum* culture grown on potato dextrose medium (bar 0.5 cm).

**Table 1 plants-12-03373-t001:** Number of somatic embryos for various *Fusarium oxysporum* treatments.

Treatment	Number of Somatic Embryos/Culture
T0	82.53 ± 1.074 d
T1	94.36 ± 0.899 b
T2	102.69 ± 0.835 a
T3	84.78 ± 0.868 c
T4	85.14 ± 0.945 c

The different *F. oxysporum* levels used were a control (T0), 0.05% (T1), 0.15% (T2), 0.25% (T3), and 0.35% (T4). The MS was added with the addition of 6.72 μM of BA and 5.37 μM of NAA. The data were scored after four weeks of culture, and the values are the means ± standard errors of three replicates. Means with the same letters are not significantly different at *p* ≤ 0.05, according to DMRT.

**Table 2 plants-12-03373-t002:** The germination of somatic embryos in *Fusarium oxysporum*-elicitated treatments.

Treatment	Germination (%)	Shoot Length (mm)	Root Length (mm)
T0	38.56 ±1.87 c	3.36 ± 0.30 c	4.34 ± 0.29 b
T1	47.58 ±2.60 b	8.96 ± 0.39 b	4.90 ± 0.23 b
T2	56.63 ±1.88 a	11.16 ± 0.29 a	6.67 ± 0.30 a
T3	16.76 ±1.92 d	1.94 ± 0.22 d	2.12 ± 0.28 c
T4	12.7 ±1.91 e	1.24 ± 0.21 d	2.07 ± 0.31 c

The different *F. oxysporum* levels used were a control (T0), 0.05% (T1), 0.15% (T2), 0.25% (T3), and 0.35% (T4). The MS was supplemented with 2.24 μM of BA. The data were scored after four weeks of culture, and the values are the means ± standard errors of three replicates. Within each column, the means with the letters are not significantly different at *p* ≤ 0.05, according to DMRT.

**Table 3 plants-12-03373-t003:** Vinblastine contents (μg g^−1^ DW) for the different stages of the embryos in *Fusarium oxysporum*-elicitated treatments.

Treatment	Induction	Proliferation	Maturation	Germination
T0	0.422 ± 0.010 c	0.401 ± 0.001 b	0.788 ± 0.005 c	0.835 ± 0.012 c
T1	0.429 ± 0.011 b	0.406 ± 0.0006 b	0.813 ± 0.0007 b	0.861 ± 0.009 b
T2	0.451 ± 0.007 a	0.417 ± 0.0009 a	0.839 ± 0.002 a	0.886 ± 0.011 a
T3	0.408 ± 0.013 d	0.395 ± 0.001 c	0.775 ± 0.004 d	0.827 ± 0.010 d
T4	0.415 ± 0.011 d	0.392 ± 0.003 c	0.771 ± 0.001 d	0.823 ± 0.018 d

The different *F. oxysporum* levels used were a control (T0), 0.05% (T1), 0.15% (T2), 0.25% (T3), and 0.35% (T4). The MS was supplemented with 2.60 μM of gibberellic acid (GA_3_). The data were scored after 45 days of culture. The values are the means ± standard errors of three replicates. Within each column, the means with the letters are not significantly different at *p* ≤ 0.05, according to DMRT.

**Table 4 plants-12-03373-t004:** Vincristine contents (μg g^−1^ DW) in the different stages of the embryos in *Fusarium oxysporum*-elicitated treatments.

Treatment	Induction	Proliferation	Maturation	Germination
T0	0.083 ± 0.014 c	0.185 ± 0.011 c	0.181 ± 0.002 b	0.254 ± 0.007 c
T1	0.088 ± 0.011 b	0.191 ± 0.008 b	0.184 ± 0.001 b	0.275 ± 0.011 b
T2	0.095 ± 0.010 a	0.199 ± 0.012 a	0.192 ± 0.0007 a	0.307 ± 0.016 a
T3	0.076 ± 0.011 d	0.182 ± 0.013 c	0.170 ± 0.002 c	0.242 ± 0.013 d
T4	0.074 ± 0.009 d	0.177 ± 0.011 d	0.168 ± 0.0006 c	0.239 ± 0.008 d

The different *F. oxysporum* levels used were the control (T0), 0.05% (T1), 0.15% (T2), 0.25% (T3), and 0.35% (T4). The MS was supplemented with 2.24 μM of BA. The data were scored after 45 days of culture. The values are the means ± standard errors of three replicates. Within each column, the means with the letters are not significantly different at *p* ≤ 0.05, according to DMRT.

**Table 5 plants-12-03373-t005:** Sugar, protein, and proline contents (mg g^−1^ FW) during the maturation stage of the embryos in a *Fusarium oxysporum*-treated culture.

Treatment	Sugar	Protein	Proline
T0	16.475 ± 0.009 d	4.517 ± 0.018 d	6.692 ± 0.010 d
T1	18.957 ± 0.011 b	5.084 ± 0.011 b	7.745 ± 0.011 b
T2	21.663 ± 0.010 a	5.378 ± 0.013 a	8.255 ± 0.009 a
T3	17.434 ± 0.009 c	4.657 ± 0.019 c	6.947 ± 0.010 c
T4	17.785 ± 0.006 c	4.695 ± 0.016 c	7.016 ± 0.008 c

The different *F. oxysporum* levels used were a control (T0), 0.05% (T1), 0.15% (T2), 0.25% (T3), and 0.35% (T4). The MS medium was supplemented with 2.60 μM of gibberellic acid (GA_3_). The data were scored after 30 days of culture. The values are the means ± standard errors of three replicates. Within each column, the means with the letters are not significantly different at *p* ≤ 0.05, according to DMRT.

**Table 6 plants-12-03373-t006:** Sugar, protein, and proline contents (mg g^−1^ FW) during the germination stage of the embryos in a *Fusarium oxysporum*-treated culture.

Treatment	Sugar	Protein	Proline
T0	12.355 ± 0.011 d	4.675 ± 0.019 d	5.847 ± 0.010 d
T1	13.282 ± 0.008 b	5.116 ± 0.017 b	6.696 ± 0.008 b
T2	14.967 ± 0.010 a	5.457 ± 0.011 a	7.254 ± 0.009 a
T3	12.674 ± 0.011 c	4.817 ± 0.014 c	6.065 ± 0.008 c
T4	12.742 ± 0.009 c	4.863 ± 0.018 c	6.146 ± 0.010 c

The different *F. oxysporum* levels used were a control (T0), 0.05% (T1), 0.15% (T2), 0.25% (T3), and 0.35% (T4). The MS medium was supplemented with 2.24 μM of BA. The data were scored after 30 days of culture. The values are the means ± standard errors of three replicates. Within each column, the means with the letters are not significantly different at *p* ≤ 0.05, according to DMRT.

## Data Availability

All data are presented in the article.

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
