# Peer review of "Fungal Elicitation Enhances Vincristine and Vinblastine Yield in the Embryogenic Tissues of Catharanthus roseus"

_plants, 2023, doi:10.3390/plants12193373_

Round 1

Reviewer 1 Report

Comment 01:

The article titled "Effect of Fusarium oxysporum on vinblastine and vincristine alkaloid yield in Catharanthus roseus embryo cultures," submitted to the journal "Plants" under the manuscript ID plants-2609544, is an explorative foray into the domain of fungal elicitation and its impact on medicinal plant cultures. This research, categorized as an article, proffers an in-depth analysis of the potential for Fusarium oxysporum to enhance the yield of vinblastine and vincristine alkaloids in Catharanthus roseus embryo cultures.

From an academic purview, this endeavor, as detailed in the provided abstract, strives to extend our understanding of fungal elicitation mechanisms in plant in vitro cultures. It delves into the intricacies of cellular stress responses, metabolic shifts, and resultant differential alkaloid production, rendered discernible through methodical treatment concentration gradations.

Socio-economically, the implications of this research are profound. Vinblastine and vincristine are pivotal in the chemotherapy regimen, and any advancement in their yield optimization could significantly augment global healthcare outcomes by potentially rendering cancer treatments more accessible and affordable.

Comment 02:

While this initial abstract provides a promising glimpse, a thorough evaluation of the full article is essential to critically appraise methodological rigor, data interpretation, and any inherent biases. It would also be pertinent to scrutinize the reproducibility of the findings, long-term ramifications on plant health due to fungal elicitation, and delve into the molecular dynamics underpinning the observed results.

Comment 03:

At certain junctures, the presented results appear to be predominantly descriptive, potentially missing deeper analytical insights that could elucidate underlying correlations and patterns. For a more comprehensive understanding and to elevate the academic robustness of this study, it is recommended that the authors engage in a meticulous correlation analysis. Additionally, a Principal Component Analysis (PCA) could offer multidimensional insights, allowing for the discernment of principal factors and the relationships amongst variables. Such analytical enhancements would undoubtedly fortify the study's conclusions and its overall contribution to the field.

Comment 04:

While the tabular representation of results provides a structured overview, it may not sufficiently highlight or visually accentuate certain intriguing findings embedded within the data. Tables, though informative, often require the reader to meticulously dissect the information, which can sometimes lead to overlooking nuanced variations or patterns.

To better showcase these results and offer a more intuitive grasp of the data's significance, it would be beneficial to integrate graphical representations. Box plots, for instance, can succinctly depict data distribution, median values, and potential outliers, providing a snapshot of variability across different treatments or conditions. Similarly, histograms can effectively illustrate frequency distributions, enabling readers to discern predominant trends or patterns swiftly.

Incorporating such graphical tools would not only enhance the visual appeal of your presentation but also ensure that pivotal findings are immediately discernible, facilitating a more impactful conveyance of the study's implications.

Comment 05:

The study delves into the activity of enzymes like SOD, CAT, and APX, which are integral to understanding a plant's response to oxidative stress. However, the absence of GPX (Glutathione Peroxidase) is noteworthy. GPX plays a significant role in detoxifying reactive oxygen species and works in tandem with the enzymes already examined. Incorporating GPX could provide a rounded view of the antioxidative defenses. It would be beneficial to understand the reasoning behind its omission.

Comment 06:

I would like to emphasize that the research presented is undoubtedly intriguing and holds substantial promise for the scientific community. The critiques provided above are intended not as a discrediting measure but rather as constructive feedback aimed at refining and enhancing the overall quality and impact of the work. 

Upon evaluating the manuscript, it is evident that it has been methodically crafted. The narrative is coherent and largely well-structured, which contributes to a clear conveyance of the research insights. The overall quality of writing in the manuscript is commendable.

Author Response

Author’s responses

Reviewer # 1

Thank you very much for your encouraging and exciting words about our work.

Yes, we accept that the current observation is rigorous and a bit descriptive. Here, the fungus extract was added to the media, and the yield of vincristine and vinblastine was quantified. During quantative analysis the defense response behavior of plant cell was monitored. Our focus was to measure the yield only; we avoided unwanted factors, analyses and interventions.

Along with tabular representation of results, graphical / histograms based data are now provided as per reviewer #1 instruction. This is a good piece of suggestion.

The enzymes SOD, CAT, and APX activity were studied to understand the process of oxidative stress. There are other antioxidant enzymes (like GPX, MDH etc.) and non-enzymatic compounds are also used for studying plant defense responses. We selected the first three omitting the others. Inclusion of all would have been more exhaustive.

All other needed minor corrections are made and highlighted.

Prof. (Dr) A. Mujib

Reviewer 2 Report

Reviewer Reports:

I recommend a minor amendment at this level.

General comments:

The manuscript entitled “Fungal Elicitation Enhances Vincristine and Vinblastine Yield in Catharanthus roseus Embryogenic Tissues” was reviewed. The work carried out in the manuscript is interesting and aimed at using Fusarium oxysporum fungus as a biotic elicitor and the yield of vinblastine and vincristine was measured in cultures. Better connect your research findings to previous works published in Plants and in other top journals. Please also remove ANY lumped references. Please define each of them separately to avoid inappropriate citations. It is recommended that the authors work with a science editor who is proficient in the Native English language to improve the organization and delivery of some portions of the manuscript. Please provide a graphical abstract to provide a visual summary of the main findings of the study. The journal's author guidelines and instructions should be followed in preparing the revised version. Other main remarks that in my opinion need attention are the following:

Detailed comments:

In the abstract, please add an indication of the achievements from your study that are relevant to the journal's scope. Please be concise - maximum 1-2 lines.

The review of the literature needs more updating with works to have a clear and concise state-of-the-art analysis. This should more clearly show the knowledge gaps identified and link them to the paper's goals. The introduction section is poorly organized. While the general introduction is acceptable, the state-of-the-art review that follows is very difficult to understand and no specific thoughts can be inferred. The major defect of this study is the debate or argument is not clearly stated. You may see these articles and follow them in the revised version. The relevant reference may be of interest to the author according to below: https://www.mdpi.com/2071-1050/14/19/12942; https://doi.org/10.1016/j.eti.2021.101357 ;Please eliminate the use of redundant words. Eg. In this way, Recently, Respectively, therefore, currently, thus, hence, finally, to do this, first, in order, however, moreover, nowadays, today, consequently, in addition, additionally, furthermore. Please revise all similar cases, as removing these term(s) would not significantly affect the meaning of the sentence. This will keep the manuscript as CONCISE as possible. Please check ALL. Avoid beginning or ending a sentence with one or a few words, they are usually redundant. Kindly revise all.

Please avoid having one heading after another with no discussion in between as in the case of Sections 2 and 2.1. Kindly inspect the entire document for similar instances and revise accordingly. Please add in the beginning of your scientific hypothesis. In the course of describing the performed actions, please provide reader guidance, sufficient for understanding why those actions have been performed. The percentage purity and company of all reagents/chemicals utilized must be reported.

The structure of this work should be reorganized. For example, the Section of results should be combined with the Discussion. The authors are suggested to have the results and discussion part together. All the findings of the current work need to be compared and discussed with the results of other researchers finding instead of having a general comparison with other researchers' works. The authors should perform a comparison between the forecasting results. In your discussion section, please link your empirical results with a broader and deeper literature review. Could you summarize the key findings of the paper regarding the impact of fungal elicitation on vincristine and vinblastine yield in Catharanthus roseus? Were there any notable changes in the secondary metabolite profile of Catharanthus roseus following fungal elicitation? Can you explain the potential mechanisms or pathways by which fungal elicitation may enhance the production of vincristine and vinblastine in Catharanthus roseus?

Please make sure your conclusions section underscores the scientific value-added of your paper, and/or the applicability of your findings/results. Highlights the novelty of your study. In the conclusions, in addition to summarising the actions taken and results, please strengthen the explanation of their significance. It is recommended to use quantitative reasoning compared with appropriate benchmarks, especially those stemming from previous work. Did the paper discuss any practical applications or implications of these findings for the pharmaceutical or agricultural industries? Were there any limitations or challenges encountered during the study that might affect the broader applicability of the results? What future research directions or experiments does the paper suggest in order to further explore the use of fungal elicitation for improving alkaloid production in medicinal plants like Catharanthus roseus?

Please check the reference section carefully and correct the inconsistency. Please update this section.

Moderate improvement is necessary

Author Response

Author’s responses

Reviewer # 2

Introduction section is now improved; new state of the art literatures have been added. The suggested articles and references (https://www.mdpi.com/2071-1050/14/19/12942; https://doi.org/10.1016/j.eti.2021.101357) by reviewer #2 do not match with this present study and the subject topics are totally different. I am afraid of inclusion.

The indicated redundant words are eliminated or minimized without affecting the meaning of the sentence.

The sections heading 2 and 2.1, and other area of similar nature have been corrected.

The results and discussion are two very components of an article. The observation of this article is quite big, containing tables, figures and photoplates in good numbers. These are elaborately explained in discussion with other authors’ view, debate and argument. We purposefully kept these two sections separate and are a common practice, which will add more value to an article.

The possible mechanisms by which fungal elicitation improve the yield of alkaloids in Catharanthus roseus is already in place in discussion section. The reviewer might have overlooked.

As per reviewer #2 suggestion, the conclusion part has been modified highlighting the achievement.

Reference section is checked carefully and the inconsistency, if any, is removed.

Prof. (Dr). A. Mujib

Round 2

Reviewer 1 Report

Dear Authors,

I recently had the opportunity to review your manuscript titled "Fungal Elicitation Enhances Vincristine and Vinblastine Yield in Catharanthus roseus Embryogenic Tissues" (I.D. plants-2609544). Your study certainly presents elements of interest and novelty. After carefully reviewing the updated version, I appreciate the revisions and additions you've made.

I would like to thank you for taking my suggestions into account.

Kind regards,

A Reviewer 1